# A New Hydrometallurgical Process for Metal Extraction from Electric Arc Furnace Dust Using Ionic Liquids

**DOI:** 10.3390/ma15238648

**Published:** 2022-12-04

**Authors:** Samaneh Teimouri, Johannes Herman Potgieter, Mari Lundström, Caren Billing, Benjamin P. Wilson

**Affiliations:** 1Sustainable and Innovative Minerals and Metals Extraction Technology (SIMMET) Group, School of Chemical and Metallurgical Engineering, University of the Witwatersrand, Private Bag X3, Wits 2050, South Africa; 2Department of Natural Science, Manchester Metropolitan University, Chester Street, Manchester M1 5GD, UK; 3Hydrometallurgy and Corrosion, Department of Chemical and Metallurgical Engineering (CMET), School of Chemical Engineering, Aalto University, P.O. Box 16200, FI-00076 Espoo, Finland; 4Molecular Sciences Institute, School of Chemistry, University of the Witwatersrand, Private Bag X3, Wits 2050, South Africa

**Keywords:** ionic liquid [Bmim^+^HSO_4_^−^], electric arc furnace dust (EAFD), kinetics, metal extraction, XRD, SEM-EDS

## Abstract

This research proposes a new hydrometallurgical method for Zn, In, and Ga extraction, along with Fe as a common impurity, from electric arc furnace dust (EAFD), using ionic liquids. EAFD is a metal-containing waste fraction generated in significant amounts during the process of steelmaking from scrap material in an electric arc furnace. With valuable metal recovery as the main goal, two ionic liquids, [Bmim^+^HSO_4_^−^] and [Bmim^+^Cl^−^], were studied in conjunction with three oxidants: Fe_2_(SO_4_)_3_, KMnO_4_, and H_2_O_2_. The results indicated that the best combination was [Bmim^+^HSO_4_^−^] with [Fe_2_(SO_4_)_3_]. An experimental series subsequently demonstrated that the combination of 30% *v*/*v* [Bmim^+^HSO_4_^−^], 1 g of [Fe_2_(SO_4_)_3_], S/L ratio = 1/20, a 240 min leaching time, and a temperature of 85 °C was optimal, resulting in maximum extractions of 92.7% Zn, 97.4% In, and 17.03% Ga. In addition, 80.2% of the impurity metal Fe was dissolved. The dissolution kinetics of these four elements over a temperature range of 55–85 °C was found to be diffusion controlled. The remaining phases present in the leached residue were low amounts of ZnO, Fe_3_O_4_, ZnFe_2_O_4_, and traces of Ca(OH)_2_ and MnO_2_, and additional sharp peaks indicative of PbSO_4_ and CaSO_4_ appeared within the XRD pattern. The intensity of the peaks related to ZnO and Fe_3_O_4_ were observed to have decreased considerably during leaching, whereas some of the refractory ZnFe_2_O_4_ phase remained. SEM-EDS analysis revealed that the initial EAFD morphology was composed of spherical-shaped fine-grained particle agglomerates, whereas the leached residue was dominated by calcium sulphate (Ca(SO_4_))-rich needle-shaped crystals. The results clearly demonstrate that [Bmim^+^HSO_4_^−^] is able to extract the target metals due to its acidic properties.

## 1. Introduction

The global demand for critical and valuable metals is growing rapidly due to their wide-ranging applications in various modern electronic devices and is consequently beginning to outstrip their current supply and production [1]. The global metal industry is therefore pursuing novel, efficient routes to recover critical and valuable metals from not only different primary sources but also secondary and waste resources, including electric arc furnace dust (EAFD), zinc electroplating anode slime, and electrical/electronic equipment (WEEE) to fulfil the heightened demand. As part of this shift, there is greater emphasis on the use of energy-efficient processes that are environmentally benign and improve material circularity [2].

The predominant process used for recycling end-of-life iron-based products (secondary resource) into steel is the electric arc furnace (EAF) [3,4]. During this process, the furnace temperature can reach 1600 °C or higher, at which many of the elements within iron scraps, such as Zn, Fe, Cr, Mn, and Pb, can be volatilised. When the furnace cools down, this vapour phase produces a large amount of unwanted powder known as electric arc furnace dust (EAFD) [5]. Currently, steel production from secondary resources is >600 M tons per year globally, and it has been estimated that an EAF generates approximately 20 kg of dust per ton of steel produced from iron scraps [4]. Owing to the elemental content, the EAFD produced during the steelmaking process is simultaneously an environmental pollutant if deposited on land—due to containing Pb and Cr—and a valuable industrial waste because of the substantial amounts of useful metals, such as Cu, Ni, Zn, and Fe, which can be recovered and returned to the steel production process [6].

Pyrometallurgical and hydrometallurgical processes have been investigated to recover valuable metals from EAFD. The pyrometallurgical Waelz process is a dominant technique used industrially to recover over 80% of steel dust [7]. Rütten (2008) investigated the Waelz process and recovered about 90% Zn. He used coke (silica and/or lime) as the reducing agent, through which the carbothermal reduction of Zn and Fe oxides occurred in a rotary kiln at a temperature of 1200 °C [8]. Hydrometallurgical-type processes are also considered to be appropriate for the extraction of valuable metals—even at low concentrations—from waste and secondary resources, such as EAFD. Interestingly, although Paul Walden [9] synthesised the first ionic liquid (ethyl ammonium nitrate [EtNH_3_NO_3_]) as early as 1914, it was only towards the end of the twentieth century that ionic liquids started to gain prominence among researchers from different disciplines, including hydrometallurgy, as potential novel solvents for the extraction and recovery of valuable and critical metals [9,10].

Traditionally, in the scientific literature, acidic solutions of HCl, H_2_SO_4_, and HNO_3_ have been used for the hydrometallurgical extraction of target metals such as Zn from EAFD [5,6,11,12,13,14,15]. For example, Kukurugya et al. (2015) investigated the extraction and behaviour of Zn, Fe, and Ca from EAFD using a H_2_SO_4_ solution to determine the influence of the sulphuric acid concentration, liquid-to-solid (L:S) ratio, temperature, and leaching time. Their research found that under optimal conditions of 80 °C with 1 M H_2_SO_4_ and an L:S ratio of 50, a maximum Zn extraction of 87% was achieved. Due to the fast dissolution of Zn, the maximum feasible extraction of Zn into the leaching solution could be accomplished within the first few minutes of leaching, eliminating extended leaching times [6]. The extraction of Zn and Fe from EAFD with H_2_SO_4_ has also been investigated by Oustadakis et al. (2010). In this work, the influence of parameters such as the H_2_SO_4_ concentration, temperature, and solid-to-liquid (S/L) ratio on the efficiency of metal extraction were examined by the statistical analysis of experiments (factorial design). It was found that the highest Zn extraction of 80% with simultaneous 45% Fe dissolution was achieved with 3 M H_2_SO_4_ at 60 °C and a 10% S/L ratio [5]. In contrast, Halli et al. (2018) employed the alkaline roasting of EAFD with NaOH at 450 °C, followed by organic acid leaching with 0.8 M citric acid at 40 °C for 2 h with oxygen purging. Under these conditions, the selective leaching of Zn over Fe was obtained (100% Zn vs. <10% Fe), and additionally, over 80% Pb was extracted [4].

Ionic liquids essentially belong to the molten salt family and comprise ions, typically a large organic cation and organic or inorganic anions of different sizes. The size difference between the cation and anion lowers the lattice energy, which makes them liquid at temperatures below 100 °C [11]. By careful consideration of their properties, different cations and anions can be combined to synthesise thousands of distinct ionic liquids with variable physicochemical properties based on the application of interest or intended purposes [10]. Ionic liquids may have remarkable properties, such as low melting points, negligible vapour pressure, and excellent thermal/chemical/electrochemical stability, and can potentially be recycled and reused, which has the added benefit of reducing the levels of chemical consumption as, for example, part of an industrial leaching process [16]. Recently, the use of ionic liquids as lixiviants has been studied for the extraction of valuable metals from different raw materials. For instance, Rüşen and Topçu (2017) examined the recovery of gold from copper anode slime with an acidic ionic liquid, 1-ethyl-3-methyl-imidazolium hydrogen sulphate (EmimHSO_4_). A maximum Au extraction of almost 90% was achieved when using an 80% ionic liquid concentration and an S/L ratio of 1/25 g/mL at 75 °C for 4 h [1]. The same authors (Topçu and Rusen, 2020) investigated Cu extraction from anode slime using 1-butyl-3-methylimidazolium-based ionic liquids with different anionic parts, i.e., HSO_4_^−^, Cl^−^, and BF_4_^−^. Their findings demonstrated that the most effective ionic liquid was 1-butyl-3-methylimidazolium hydrogen sulphate (BmimHSO_4_) under conditions of a 50% ionic liquid concentration, an S/L ratio of 1/20 g/mL at 50 °C, and an 8 h leaching time, which yielded almost 30% Cu extraction [9].

The recovery of In and Ga is becoming essential due to both their relative scarcity and growing demand, especially for their use in light-emitting diodes (LEDs) and other electronic devices [17]. Theocharis et al. (2021) investigated the extraction and recovery of In, Ga, Cu, Zn, and Mo from end-of-life thin-film solar panels containing CIGS (copper indium gallium selenide, CuGa_1−x_In_x_Se_2_). The CIGS photovoltaic panels were dismantled and thermally processed at 550 °C for 15 min to delaminate the ethylene vinyl acetate (EVA) and the thin layer of coated glass. Once prepared, the thermally treated samples were leached with 6 M HNO_3_ to extract the In, Ga, Cu, Zn, and Mo. The leachate solution was then subjected to solvent extraction using D_2_EHPA to separate and recover the extracted metals [2]. Zhan et al. (2015) examined an alternative method for the recovery of In and Ga from discarded light-emitting diodes (LEDs) by applying pyrolysis, physical disintegration (crushing, screening, and grinding), and vacuum metallurgy separation. The latter is a heating separation/purification process under vacuum, in which the boiling point of metals can be significantly reduced, thereby requiring less energy. These methodologies achieved In and Ga extractions of ~96% and ~94%, respectively, under optimal conditions of 1100 °C heating, 0.01–0.1 Pa vacuum pressure, and a holding time of 1 h [16].

To separate and recover Zn from a leached liquor, solvent extraction and electrowinning techniques have been applied [17,18,19,20]. For Fe recovery, the precipitation technique [21] is mostly used, and the solvent extraction process is used for In and Ga [22,23]. The EAFD residue after the required treatment to remove hazardous heavy metals, i.e., Cr and Pb, was disposed of in a landfill [24]. Recently, there has been research on the use of EAFD residue in the production of ceramics, red clay bricks [25,26,27,28], building blocks, and cement [24].

In acidic solutions (H_2_SO_4_, HNO_3_, and HCl), In can be readily extracted according to Reactions (1)–(4) [29]. The stable oxidation state of both In and Ga is +3. Since they both belong to the Boron family of elements (i.e., group 13) of the periodic table, one can extrapolate that the same/similar reactions also occur for Ga treated with these acidic solutions.
In_2_O_3_ + 6H^+^→ 2In^3+^ + 3H_2_O (general reaction in an acidic medium)(1)
In_2_O_3_ + 3H_2_SO_4_→ In_2_(SO_4_)_3_ + 3H_2_O(2)
In_2_O_3_ + 6HNO_3_→ 2In(NO_3_)_3_ + 3H_2_O(3)
In_2_O_3_ + 6HCl→ 2InCl_3_ + 3H_2_O(4)

Based on these acidic reactions, three imidazolium-based ionic liquids with similar anions to these mineral acids—[Bmim^+^HSO_4_^−^], [Bmim^+^NO_3_^−^], and [Bmim^+^Cl^−^]—were selected for this study. Figure 1 illustrates the chemical structures of the cationic part (1-butyl-3-methylimidazolium–Bmim^+^) and the different anions (Cl^−^, HSO_4_^−^, and NO_3_^−^) that compose the three distinct ionic liquids. However, as [Bmim^+^NO_3_^−^] has limited availability due to global production issues, it was excluded from this investigation.

This work aims to determine the feasibility of using imidazolium-based ionic liquids for Zn, In, and Ga extraction from EAFD, along with Fe as a common impurity. The effects of ionic liquid with the same imidazolium cation (1-butyl-3-methylimidazolium) and different anionic components (HSO_4_^−^ and Cl^−^) were examined experimentally. This involved the investigation of two ionic liquids—[Bmim^+^HSO_4_^−^] and [Bmim^+^Cl^−^]—with three oxidants (Fe_2_(SO_4_)_3_, KMnO_4_, and H_2_O_2_) to determine which ionic liquid and oxidant combination performs the best in extracting the target metals. Following the initial tests, the influence of parameters such as the concentration of the ionic liquid, oxidant concentration, solid-to-liquid (S/L) ratio, time, and temperature were optimised to achieve the maximum extraction of Zn, In, Ga, and Fe (as the impurity) from electric arc furnace dust (EAFD). A kinetic study of the dissolution of these elements from EAFD in a 30% *v*/*v* [Bmim^+^HSO_4_^−^] solution was conducted over a temperature range of 55–85 °C, and the results were evaluated using shrinking-core models. The leached residue of EAFD was also analysed by X-ray diffraction (XRD), scanning electron microscopy in conjunction with energy-dispersive X-ray spectroscopy (SEM-EDS), and particle size distribution (PSD) analysis to identify changes in the main phases and particles following leaching in the ionic liquid medium. To the authors’ knowledge, this approach has not been researched previously for EAFD and can contribute to a novel process for metal recovery from this source and increase the knowledge about the use of ionic liquids for various metal extractions from secondary resources.

## 2. Experimental Procedures

### 2.1. Materials

The electric arc furnace dust (EAFD) employed in this research was obtained from the Ovako Imatra Oy steel production plant in Finland and was used “as received”. Table 1 lists the results for the total acid leaching of a well-mixed representative EAFD sample indicating the chemical composition of the EAFD used, as determined by atomic absorption spectroscopy (AAS), inductively coupled plasma–optical emission spectrometry (ICP-OES), and sulphur/carbon analysis, adapted from Halli et al. (2017) [14]. From the analyses, it was determined that the main components within the EAFD were Zn, Fe, Ca, Mn, and Pb. Additionally, the mineralogy of the material was studied by X-ray diffraction (XRD), and the resultant pattern is presented in Figure 2. The XRD phase analysis identified the main phases in EAFD as zincite (ZnO), zinc ferrite (ZnFe_2_O_4_), and magnetite (Fe_3_O_4_), with some minor phases of Ca(OH)_2_, PbO, MnO_2_, and MgO. The mineral phase evaluation from XRD correlated well with the chemical composition determined after total acid digestion. The particle size distribution (PSD) of the EAFD sample was measured by laser diffraction, as shown in Figure 3. As shown, the frequency and cumulative distribution of EAFD particles primarily comprise two distinct size fractions: very fine grains of 0.01–0.15 µm and a portion of coarser particles of about 10–100 µm. From the PSD, it was found that 50% of the particles were below 14.2 µm, while the majority of the particles (90%) were <100 µm.

### 2.2. Apparatuses

The instrumentations used in this research were as follows: the leached solution was analysed by AAS (atomic absorption spectroscopy, Varian AA240, Palo Alto, California, USA) for the contents of In, Ga, and Fe and by ICP-OES (inductively coupled plasma–optical emission spectrometry, Perkin Elmer Optima 7100 DV, Beaconsfield, Bucks, PA, USA) for Zn, as well as a sulphur/carbon analyser (Eltra CS-580 analyser, Haan, Germany) to determine the chemical composition of the EAFD. Phase analysis was carried out by XRD (X-ray diffraction, PANalytical-X’Pert PRO Powder, Almelo, Netherlands), applying a 40 mA current and 45 kV acceleration voltage, with a CuKα radiation source. The particle size distribution of the EAFD was measured by a Mastersizer 3000 (Malvern Instruments Ltd., Malvern, UK). The pH and redox potential in the leaching solutions were measured with a digital pH electrode (HI 11,310, Hanna Instruments, Woonsocket, Smithfield, RI, USA) and InLab Ag/AgCl 3M KCl (Mettler Toledo, Columbus, OH, USA) reference electrode, respectively. The morphological examination of the EAFD particles was conducted with an SEM-EDS (scanning electron microscopy, A LEO 1450, Carl Zeiss Microscopy GmbH, Jena, Germany) in conjunction with EDS (energy-dispersive X-ray spectroscopy, Link Inca X-sight 7366, Oxfordshire, UK).

The chemicals employed in this research were two ionic liquids, 1-butyl-3 methylimidazolium hydrogen sulphate [Bmim^+^HSO_4_^−^] and 1-butyl-3 methylimidazolium chloride [Bmim^+^Cl^−^], and three oxidants, ferric sulphate (Fe_2_(SO_4_)_3_), potassium permanganate (KMnO_4_), and hydrogen peroxide (H_2_O_2_). Indium and Ga oxide (In_2_O_3_ and Ga_2_O_3_) were also added to the EAFD sample in order to ascertain the possibility of In and Ga recovery from this type of matrix. All of the chemicals were purchased from Merck/Sigma-Aldrich, Finland, and related solutions were made using ultrapure water (>18 MΩ-cm, Millipore Milli-Q, Merck, Espoo, Finland).

### 2.3. Procedures

The leaching of EAFD with ionic liquid as the primary lixiviant was conducted under atmospheric pressure in a 250 mL three-neck round-bottom reactor immersed in a water bath placed on top of a hot-plate magnetic stirrer. One neck of the reactor was attached to a condenser, one was used to accommodate a thermometer that was immersed in the leaching solution, and the third opening was used for sampling. Agitation was provided by a stirring bar within the reactor in order to suspend the particles in the ionic liquid solution and produce a uniform pulp. A typical leaching experiment was carried out: a pre-determined amount of ionic liquid solution with a known concentration was heated to the desired temperature, and then the selected oxidant and a known amount of EAFD enriched with 5% each of In_2_O_3_ and Ga_2_O_3_ were added to the preheated ionic liquid solution.

Initial experiments were performed using two ionic liquids ([Bmim^+^HSO_4_^−^] and [Bmim^+^Cl^−^]) and three oxidants to determine which combination of ionic liquid and one of the three proposed oxidants would be the most effective and lead to the highest extraction of the target metals from EAFD. The specific experimental conditions were as follows: 50 mL of ionic liquid ([Bmim^+^HSO_4_^−^] or [Bmim^+^Cl^−^]) with a concentration of 50% (*v*/*v*), oxidant concentration (one of either KMnO_4_ (0.5 g), Fe_2_(SO_4_)_3_ (0.5 g), or H_2_O_2_ 50% (1 mL)), a temperature of 65 °C, S/L = 1/20, a stirring speed of 500 rpm, and a leaching time of 8 h.

After determining the most effective ionic liquid and oxidant combination, the operating parameters, namely, 50 mL of ionic liquid with different concentrations (30, 40, 50, and 60% *v*/*v*), oxidant masses (0.25, 0.5, 0.75, 1 g for KMnO_4_ or Fe_2_(SO_4_)_3_) or oxidant volumes (1, 1.5, 2, and 2.5 mL for 50% H_2_O_2_), S/L ratios (1/10, 1/15, 1/20, and 1/25 g/mL), and temperatures (55, 65, 75, and 85 °C), were examined to determine the optimal conditions for maximum Zn, In, and Ga extraction from EAFD. To monitor the progress of the dissolution experiment, 2 mL aliquots of the leach solution were taken from the reactor using a pipette at regular time intervals between 0.5 and 8 h and analysed to determine the contents of the target metals. For the kinetic studies (carried out at 55, 65, 75, and 85 °C), 100 mL of a 30% (*v*/*v*) ionic liquid solution was prepared, and 2 mL aliquots were withdrawn from the reactor at the following intervals: 10, 20, 30, 45, 60, 90, 120, 180, and 240 min. In all cases, 2 mL of fresh ionic liquid was added to the reactor after each withdrawal to maintain the appropriate S/L ratio.

## 3. Results and Discussion

### 3.1. The Combination of Ionic Liquid and Oxidant

To determine the most effective ionic liquid and oxidant combination for the extraction of Zn, In, and Ga from EAFD, two imidazolium-based ionic liquids, [Bmim^+^HSO_4_^−^] and [Bmim^+^Cl^−^] (50 mL, 50% *v*/*v*), were mixed with one of three selected oxidants, KMnO_4_ (0.5 g), Fe_2_(SO_4_)_3_ (0.5 g), or 50% H_2_O_2_ (1 mL) (Figure 4). After conducting the experiments, it was found that [Bmim^+^HSO_4_^−^] performed considerably better than [Bmim^+^Cl^−^] due to the ability of [Bmim^+^HSO_4_^−^] to act as a Brønsted acid, which allows the anionic species [HSO_4_^−^] to release the associated proton [H^+^] into the solution [16,17,18]. Furthermore, the sulphate [SO_4_^2−^] is able to act as a better ligand (compared to Cl^−^) to form a stable complex with the target metals. Equation (5) shows this dissociation of [Bmim^+^HSO_4_^−^] in an aqueous solution to release [H^+^].
[Bmim^+^HSO_4_^−^] → [Bmim^+^] + [H^+^] + [SO_4_^2−^](5)

Therefore, one of the main characteristics of [Bmim^+^HSO_4_^−^] is its ability to substantially lower the leaching solution’s pH, making the solution acidic [16,17]. For instance, it was found that for 50% *v*/*v* [Bmim^+^HSO_4_^−^], the pH was ~0.88, whereas for the equivalent 50% *v*/*v* [Bmim^+^Cl^−^], the leaching solution’s pH was ~4.7. As the results in Figure 4 show, [Bmim^+^HSO_4_^−^] with the addition of any one of the three selected oxidants led to a satisfactory extraction, with Fe_2_(SO_4_)_3_ being the best oxidant. Among the three studied oxidants, Fe_2_(SO_4_)_3_ was selected due to being cheaper and safer to handle, in addition to achieving slightly more metal extraction. For [Bmim^+^HSO_4_^−^], the extraction of Zn and In was the highest, followed by the typical EAFD impurity Fe. In contrast, irrespective of the ionic liquid and oxidant combination used, the Ga extraction remained low. [Bmim^+^Cl^−^] combined with the studied oxidants resulted in the low extraction of the target metals. Consequently, the combination of [Bmim^+^HSO_4_^−^] with Fe_2_(SO_4_)_3_ was selected as the most suitable for the optimisation in the rest of the experiments.

### 3.2. The Effect of Ionic Liquid Concentration

When it comes to leaching, the concentration of the ionic liquid [Bmim^+^HSO_4_^−^] plays a significant role due to its high viscosity. In its pure state, [Bmim^+^HSO_4_^−^] is very viscous (900 mPa) [30]. However, when mixed with water, there is a noticeable reduction in viscosity due to its acidic properties, as it can release protons into the aqueous phase [16,17,31]. To find the optimal concentration of [Bmim^+^HSO_4_^−^] required for metal extraction from EAFD, specific amounts of [Bmim^+^HSO_4_^−^] were mixed with deionised water to prepare different ionic liquid concentrations of 30, 40, 50, and 60% *v*/*v*. Experiments were carried out under the following conditions: 0.5 g of oxidant Fe_2_(SO_4_)_3_, an S/L ratio of 1/20, a stirring speed of 500 rpm, and a temperature of 65 °C, and 2 mL leachate aliquots were withdrawn at regular time intervals over an 8 h period. Each leachate sample was then filtered and analysed with AAS and ICP to determine the amounts of extracted Fe, In, Ga, and Zn.

As can be seen in Figure 5, at concentrations of 50% and 60% [Bmim^+^HSO_4_^−^], the levels of extracted metals were lower than when the ionic liquid concentration was 30% and 40%. These results suggest that at higher concentrations (50% and 60% *v*/*v*) of [Bmim^+^HSO_4_^−^], the viscosity of the leaching solution is high enough to act as a barrier to the free movement and collision of ions, leading to less extraction. At the lower [Bmim^+^HSO_4_^−^] concentrations of 30% and 40% *v*/*v*, the extraction of target metals is possibly increased as a function of the balance between the reduced viscosity and the applicable concentration of the ionic liquid, in addition to the proper acidity that it provides, which gives rise to more comparable results. Of the elements of interest, Zn has the highest extraction, reaching 90% with a 40% *v*/*v* [Bmim^+^HSO_4_^−^] solution. This is due to the high solubility of ZnO in acidic solutions [6]. After Zn, In has the next highest extraction of 84%, followed by Fe with 70% under the same conditions. Conversely, Ga was found to have a low extraction level (<20%), irrespective of the [Bmim^+^HSO_4_^−^] concentration used. The extraction efficiency of the metals with 30% *v*/*v* [Bmim^+^HSO_4_^−^] was not significantly different to that with the 40% *v*/*v* [Bmim^+^HSO_4_^−^] solution. Consequently, in order to make any future extraction methodologies more economically viable, 30% *v*/*v* [Bmim^+^HSO_4_^−^] was selected as the optimal concentration and was used in further investigations.

An aqueous solution of [Bmim^+^HSO_4_^−^] containing [H^+^] and [SO_4_^2−^] displays similar properties to a sulphuric acid solution (H_2_SO_4_) [16,18]. Similar to a leaching process using sulphuric acid, the ionic liquid [Bmim^+^HSO_4_^−^] solution can also be recycled and reused in the leaching process, which can justify its initial higher cost. Bearing this in mind, Reactions (6)–(8) are proposed to occur during the extraction of zinc and iron from their main sources, ZnO, ZnFe_2_O_4_, and Fe_3_O_4_, from EAFD in an acidic solution [6]:ZnO + [SO_4_^2−^] + 2[H^+^] → ZnSO_4_ + H_2_O(6)
ZnFe_2_O_4_ + 4[SO_4_^2−^] + 8[H^+^] → ZnSO_4_ + Fe_2_(SO_4_)_3_ + 4H_2_O(7)
Fe_3_O_4_ + 4[SO_4_^2−^] + 8[H^+^] → FeSO_4_ + Fe_2_(SO_4_)_3_ + 4H_2_O(8)

### 3.3. The Effect of Oxidant Concentration

To examine the effect of the oxidant concentration on the extraction of Zn, In, Ga, and Fe (impurity) from EAFD, different amounts of Fe_2_(SO_4_)_3_ (0.25, 0.5, 0.75, and 1 g) were added to 30% *v*/*v* [Bmim^+^HSO_4_^−^] as the leaching solution. As the results in Figure 6 indicate, increasing the amount of the oxidant improved the extraction of the metals of interest. This can be expected since the availability of higher concentrations of the oxidant would provide a stronger oxidative power to the leaching solution for metal extraction. The extraction of Fe improved from around 48% to 74% when the amount of the oxidant was increased from 0.25 g to 1 g. Although Zn was previously found to have a high level of extraction (Figure 6), further minor increases in extraction to over 90% were measured with a 1 g addition of the oxidant. The extraction of In was also enhanced to 87%, whereas Ga extraction reached 18.7% with an increase to 1 g of Fe_2_(SO_4_)_3_.

### 3.4. Effect of Solid-to-Liquid (S/L) Ratio

In a leaching process, which typically involves a heterogeneous solid–liquid reaction, the correct ratio of solid to liquid (S/L) is important. To determine the optimal ratio, different amounts of EAFD were added to 50 mL of 30% *v*/*v* [Bmim^+^HSO_4_^−^] to make the following S/L ratios: 1/25, 1/20, 1/15, and 1/10. It was observed that the different S/L ratios had slightly different effects on the extraction of the studied elements (Figure 7). For Fe, a ratio of 1/15 resulted in over 73% extraction. In the case of Zn, higher extraction occurred when there was more solid (EAFD) available in the leaching solution [6]; therefore, a ratio of 1/10 provided the best results for Zn, with 91% extraction. In the case of In and Ga, a ratio of 1/25 was ideal, with about 80% In and 17% Ga extracted. Depending on the mineralogy of materials, a high amount of solid in the leaching solution can sometimes contribute to ineffective mixing within the solution. This is due to the higher-density mixture causing ineffective dispersion and mass transfer within the mixture [32,33,34,35]. With a high amount of solid in the mixture, the amount of the ionic liquid solution can also be a limiting reagent in the dissolution reaction. From Figure 7, to compromise between the different trends in leaching with varying S/L ratios, an S/L ratio of 1/20 was selected as the optimal compromise for the remainder of the experiments since it produced higher In and Ga extraction, as well as reasonable Zn and Fe extraction. It was also observed that no significant further extraction of the elements of interest takes place beyond 4 h of leaching. Therefore, the following experiments at different temperatures were conducted for up to 4 h (240 min) of total leaching time.

### 3.5. Effect of Temperature

The influence of temperature on the leaching of the target metals from EAFD in 30% *v*/*v* [Bmim^+^HSO_4_^−^] was examined by varying the temperature from 55 to 85 °C. Generally, when increasing the temperature, the extraction of the metals improved due to enhanced diffusion and the increased rate of the reaction [17,20,21,32]. Results displaying the extraction of Fe, Zn, In, and Ga at regular time intervals of 10, 20, 30, 45, 60, 90, 120, 180, and 240 min are presented in Figure 8. The influence of increasing temperature is more significant for Fe than for Zn; hence, there is a noticeable improvement in Fe extraction from 55.5% at 55 °C to 80.2% at 85 °C. Although Zn extraction was initially high (77.6% at 55 °C), there was still a noticeable increase to 92.7% when the temperature was raised to 85 °C. Indium extraction improved significantly from 67.5% to 80.9% with a change in temperature from 55 to 65 °C. With a further temperature increase to 85 °C, In extraction reached over 97%. There was also a discernible improvement in Ga extraction to almost 18% at the maximum studied temperature of 85 °C. These results show that the extraction of the target metals increases significantly in the first hour, after which it becomes more moderate with extended leaching time.

### 3.6. Kinetic Study

To study the kinetics of Fe, Zn, In, and Ga dissolution in the 30% *v*/*v* [Bmim^+^HSO_4_^−^] solution with the oxidant Fe_2_(SO_4_)_3_ (1 g), the experimental results from varying the temperature from 55 to 85 °C were evaluated with the shrinking-core model. The functions from this model considered the surface chemical reaction [(1 − (1 − X)^1/3^ = k.t], diffusion through the product [1 − 3(1 − X)^2/3^ + 2(1 − X) = k.t], and diffusion through the liquid film [X = k.t], where k is the apparent rate constant, and X is the concentration of the leachate from the leached solution. Each function was plotted over time (min), and the graph that produced straight lines with correlation coefficients (R^2^) close to 1 was considered suitable for the fitting of the experimental results and estimating the kinetic model controlling the dissolution reaction [36]. Accordingly, it was found that the kinetics of the dissolution of Fe, Zn, In, and Ga were best described by diffusion through the product. This suggested kinetic model agrees well with a previous study that also followed a diffusion-controlled model [6].

Figure 9 illustrates the fitness of the diffusion equation over time for Fe, displaying straight lines for each studied temperature. The graph clearly shows that the dissolution reaction was faster with a steeper slope in the first hour of leaching before becoming more moderate. The reason for this could be that more EAFD and oxidizing reagent, which helps leaching, were accessible to extract Fe from the easily extractable magnetite (Fe_3_O_4_) before they gradually decreased such that Fe needed to be subsequently extracted from a refractory phase such as zinc ferrite (ZnFe_2_O_4_), which is a slower and/or more difficult process and which reduces the rate of extraction.

By applying the Arrhenius equation, k = Ae^−Ea/RT^, from the plot of *ln* k vs. the inverse of the studied temperatures (1000/T), the activation energy (E_a_) for each part of the reaction can be obtained from the slope in Figure 10 (slope = −E_a_/R). As expected, E_a_ for the initial part of the curve—which comprises Fe extraction from a more easily extractable phase (Fe_3_O_4_)—was lower than the second portion, which is related to Fe extraction from a phase that is more chemically resistant (ZnFe_2_O_4_), with calculated values of 20.8 and 34.6 kJ/mol respectively.

As for Fe, the kinetic process of Zn extraction was also governed by diffusion. Figure 11 demonstrates the functionality of the diffusion equation over time (min), displaying linear relationships for Zn extraction at different temperatures (55 to 85 °C). In general, the extractable Zn comes primarily from the ZnO phase, with a minor contribution from the slower-dissolving refractory ZnFe_2_O_4_ phase. Towards the latter part of the reaction, i.e., after 1 h, the extracted Zn originates mostly from the ZnFe_2_O_4_ phase, as the ZnO phase has already largely dissolved [6,11].

From the Arrhenius plots (Figure 12), the obtained E_a_ for the initial part of the Zn extraction was 7.4 kJ/mol, whereas for the second part, E_a_ = 8.2 kJ/mol. These noticeably low activation energies for Zn extraction, irrespective of the Zn-containing phase, clearly explain the faster dissolution of Zn and its higher extraction yield compared to those of Fe.

Figure 13 presents the kinetic dissolution process of In in the 30% *v*/*v* [Bmim^+^HSO_4_^−^] solution controlled by the diffusion model. The process for In extraction takes place in a single step for each studied temperature, with R^2^ values close to one indicating a good fit. The activation energy determined from the plot of *ln* k vs. 1000/T was 33.6 kJ/mol (Figure 14), which is in good agreement with a diffusion-controlled process [31,36].

There was a low extraction yield of Ga throughout this work, most likely due to Ga being inherently less active or soluble in the ionic liquid medium. Nevertheless, the kinetics of Ga dissolution in the 30% *v*/*v* [Bmim^+^HSO_4_^−^] solution (Figure 15) correlated well with the diffusion-controlled model, again showing two distinct kinetic regions. The activation energy for Ga in the first part of the extraction (10–45 min) was 12.2 kJ/mol, and it was 30.3 kJ/mol in the second part, correlating with the slower Ga extraction depicted in Figure 16.

The kinetic model controlling the dissolution of target metals in a leaching system can be predicted from the magnitude of the activation energy that a process requires. Typically, activation energies below 40 kJ/mol indicate a diffusion-controlled process, while values greater than 40 kJ/mol indicate chemical reaction control [31,36,37]. Therefore, in this research, the activation energies for Fe, Zn, In, and Ga were all below 40 kJ/mol and in the range of diffusion-controlled processes. Table 2 also lists the extraction efficiency and activation energies for Zn and Fe extraction from EAFD using a 1 M H_2_SO_4_ solution at temperatures from 20 to 95 °C, adapted from work conducted by Kukurugya et al. (2015) [6], to compare to the results obtained in this work using a 30% *v*/*v* [Bmim^+^HSO_4_^−^] solution.

While the recoveries of Fe and Zn in both media are very similar, the activation energies in the ionic liquid medium are generally lower, indicating that it is potentially a more favourable medium to use for leaching. Because it is recyclable, it supports the decision to use ionic liquid [Bmim^+^HSO_4_^−^] as a leachant.

### 3.7. Analysis of the Solid Residue

#### 3.7.1. XRD Analysis

The EAFD sample (before leaching) and the residue (after leaching in 30% *v*/*v* [Bmim^+^HSO_4_^−^] solution at temperatures from 55 to 85 °C) were examined by XRD and compared to another experiment with 2 M H_2_SO_4_ at 85 °C. As can be seen in Figure 17, the EAFD before leaching mostly contains ZnO, ZnFe_2_O_4_, and Fe_3_O_4_, as well as small amounts of Ca(OH)_2_, PbO, MnO_2_, and MgO. After leaching in 30% *v*/*v* acidic ionic liquid [Bmim^+^HSO_4_^−^], the intensity of the peaks belonging to ZnO and Fe_3_O_4_ decreased considerably, indicating their dissolution, while some of the ZnFe_2_O_4_ phase, which has a refractory nature, remained. In addition, some crystalline phases, such as PbSO_4_ and CaSO_4_, appeared in the XRD pattern after leaching. It was noticed that by increasing the temperature of the extraction process, the intensity of the peaks indicating different phases, including ZnFe_2_O_4_, decreased, indicating that the further dissolution of these phases occurred. The XRD pattern for the residue in the 30% *v*/*v* [Bmim^+^HSO_4_^−^] solution was similar to that in 2 M H_2_SO_4_, showing that the acidic ionic liquid [Bmim^+^HSO_4_^−^] can act similarly to H_2_SO_4_ as a leaching solution [5,6].

#### 3.7.2. PSD Analysis

The particle size of the leached residue of EAFD was inspected using laser diffraction to determine whether the size of particles changed after leaching in the 30% *v*/*v* [Bmim^+^HSO_4_^−^] solution. Figure 18 displays the frequency and cumulative distributions of the studied EAFD before and after leaching at temperatures of 65 and 85 °C. The graph shows that the size of the particles decreased after leaching compared to the unleached sample, as well as the occurrence of smaller particles at the higher temperature (85 °C). This is indicative of the particle dissolution that takes place as the metals are extracted.

#### 3.7.3. SEM-EDS Analysis

Backscattered electron (BSE) scanning electron microscopy in conjunction with energy-dispersive spectroscopy (SEM-EDS) was utilised to acquire additional information about the morphology and mineralogical species involved in the studied EAFD. Figure 19 displays the SEM-EDS analysis of the EAFD before leaching, which provides the chemical composition (Table 3) and the coloured map of the main elements in the examined area. The main elements detected by SEM-EDS were Zn (38.5%), O (27.5%), Fe (20.4%), Ca (3.5%), and Mn (2.6%), which correspond to the main phases identified by XRD as ZnO, ZnFe_2_O_4_, Fe_3_O_4_, and traces of Ca(OH)_2_ and MnO_2_. From a morphological point of view, the EAFD has spherical-shaped fine-grained particles in an agglomerated form with some small rectangular particles. Based on the observation of the morphology and chemical content, the spherical particles have considerable amounts of Fe and Zn, whereas the rectangular particles have a high Ca content. This observation agrees with other research on leaching EAFD in sulphuric acid [5,6,10].

The EAFD residue after leaching in the 30% *v*/*v* [Bmim^+^HSO_4_^−^] solution at different temperatures (55 to 85 °C) was also examined with SEM-EDS (Figure 20). After leaching, some needle-shaped crystals appeared, which were found to be calcium sulphate Ca(SO_4_) crystals. No obvious differences appeared in the leached residue attributable to the different leaching temperatures used.

Figure 21 demonstrates a more detailed SEM-EDS analysis of the EAFD residue after leaching in the 30% *v*/*v* [Bmim^+^HSO_4_^−^] solution at 85 °C. Points 1 and 4 are directed at a bright semi-round-shaped particle mostly consisting of Pb, O, and S, which were most likely PbSO_4_ particles. Points 2 and 5 focus on the needle-shaped crystal, confirming it comprised CaSO_4_, since it had high Ca, S, and O contents. Point 3 focuses on a round-shaped particle mostly containing Fe, Zn, and O, indicating that this was most likely ZnFe_2_O_4_. Table 4 summarises the chemical composition determined at each point under examination.

Figure 22 presents a scheme for the proposed treatment process(es) of EAFD leaching showing the residue and the leached solution with the different metallic elements, as well as the recovery methods.

## 4. Conclusions

This research focused on a new hydrometallurgical technique using ionic liquids for the extraction of Zn, In, and Ga, as well as Fe as a common impurity, from EAFD, combined with oxides of In and Ga simulating industrial waste. Two ionic liquids, [Bmim^+^HSO_4_^−^] and [Bmim^+^Cl^−^], were tested with three oxidants, Fe_2_(SO_4_)_3_, KMnO_4_, and H_2_O_2_, to determine the best combination of ionic liquid and oxidant to extract the target metals.

The results showed that the optimal combination was [Bmim^+^HSO_4_^−^] and Fe_2_(SO_4_)_3_. Influential experimental parameters such as the concentration of the ionic liquid [Bmim^+^HSO_4_^−^], oxidant concentration, solid-to-liquid ratio, time, and temperature were optimised to find the optimal conditions for the maximum extraction of all of the target metals. The experimental observations were as follows:The obtained optimal extraction conditions were 30% *v*/*v* [Bmim^+^HSO_4_^−^], 1 g of [Fe_2_(SO_4_)_3_] oxidant, an S/L ratio of 1/20, and a 4 h leaching time at a temperature of 85 °C.Under these optimal conditions, the maximum extraction obtained was 92.7% Zn, with a simultaneous Fe dissolution of 80.2%If In and Ga were present in the studied raw material, up to 97.4% In and 17.03% Ga extraction could be achieved from oxide phases with 30% *v*/*v* [Bmim^+^HSO_4_^−^] as the leaching solution; hence, the process is not very successful for Ga extraction.The dissolution kinetics of the target metals in the 30% *v*/*v* [Bmim^+^HSO_4_^−^] solution from EAFD in the temperature range of 55–85 °C was diffusion-controlled.The dissolution of the desired metals was faster in the first hour of leaching, before becoming more moderate.The calculated activation energy for the dissolution reactions of all four elements investigated indicated diffusion-controlled reaction kinetic processes, as the activation energy remained below 40 kJ/mol.

The main phases of EAFD identified by XRD were ZnO, ZnFe_2_O_4_, and Fe_3_O_4_, with traces of Ca(OH)_2_ and MnO_2_. After leaching in the 30% *v*/*v* [Bmim^+^HSO_4_^−^] solution, the intensity of the ZnO and Fe_3_O_4_ peaks decreased substantially, indicating their dissolution. The peaks of the ZnFe_2_O_4_ phase, which is naturally refractory, remained, and the intensity decreased with increasing temperature, indicating further dissolution. In addition, PbSO_4_ and CaSO_4_ phases appeared in the XRD patterns.

The SEM-EDS analysis revealed that the morphology of the studied EAFD was agglomerated spherical-shaped fine-grained particles. After leaching, needle-shaped calcium sulphate (CaSO_4_) crystals appeared.PSD analysis showed that the size of the particles decreased after leaching in the 30% *v*/*v* [Bmim^+^HSO_4_^−^] solution when compared to the unleached sample. Moreover, the higher the temperature, the smaller the particles became.

The advantage of using acidic ionic-liquid-like [Bmim^+^HSO_4_^−^] as the leaching solution for metal extraction is that the ionic liquid is potentially recyclable and can be used again in the leaching process, which can reduce operational costs in the long run. As this study has shown, when compared to sulphuric acid, the ionic liquid [Bmim^+^HSO_4_^−^] achieved a similar or slightly higher extraction of the studied metals, with faster dissolution kinetics and lower activation energies.

## Figures and Tables

**Figure 1 materials-15-08648-f001:**
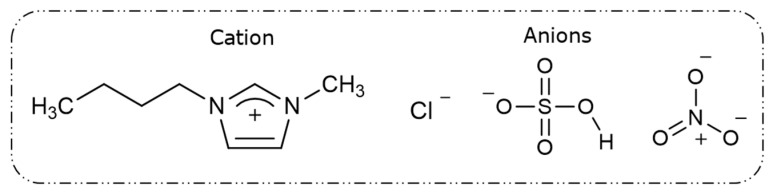
Structures of the cation (1-butyl-3-methylimidazolium–Bmim^+^) and different anions (Cl^−^, HSO_4_^−^, and NO_3_^−^) composing the ionic liquids.

**Figure 2 materials-15-08648-f002:**
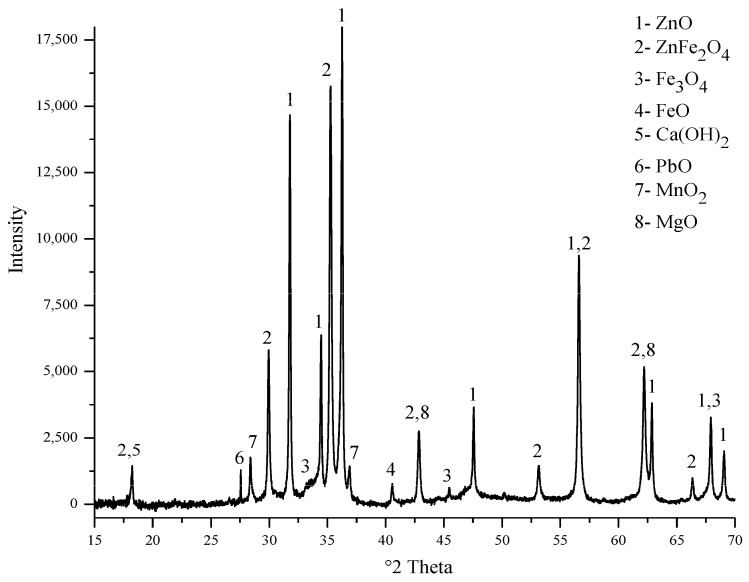
XRD pattern (mineral phase) of the EAFD.

**Figure 3 materials-15-08648-f003:**
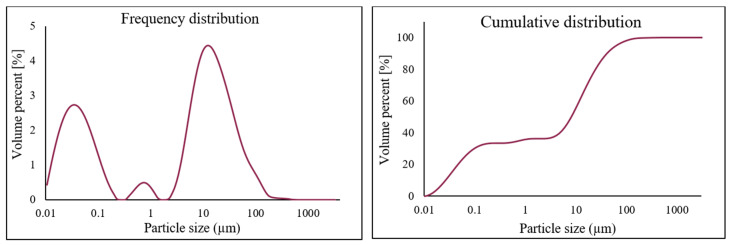
Particle size distribution (PSD) of the EAFD.

**Figure 4 materials-15-08648-f004:**
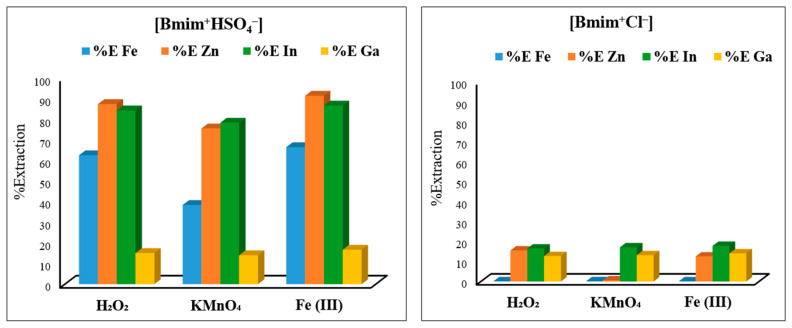
Metal extraction percentages achieved by the combination of ionic liquids [Bmim^+^HSO_4_^−^] and [Bmim^+^Cl^−^] with the three oxidants H_2_O_2_ 50% (1 mL), KMnO_4_ (0.5 g), and Fe_2_(SO_4_)_3_ (0.5 g). Experimental conditions: 50 mL of 50% (*v*/*v*) ionic liquid solution, S/L ratio of 1/20, stirring speed of 500 rpm, temperature of 65 °C, and total leaching time of 8 h.

**Figure 5 materials-15-08648-f005:**
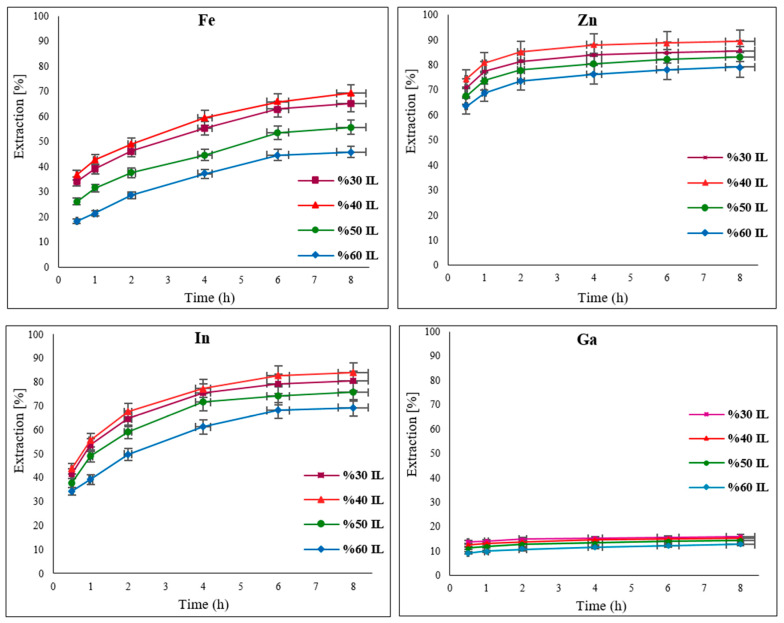
Effect of different concentrations of [Bmim^+^HSO_4_^−^] of 30, 40, 50, and 60% *v*/*v* on the extraction of Fe, Zn, In, and Ga from EAFD. Experimental conditions: 50 mL of ionic liquid solution with specified concentrations, oxidant Fe_2_(SO_4_)_3_ (0.5 g), S/L ratio of 1/20, stirring speed of 500 rpm, temperature of 65 °C, and total leaching time of 8 h.

**Figure 6 materials-15-08648-f006:**
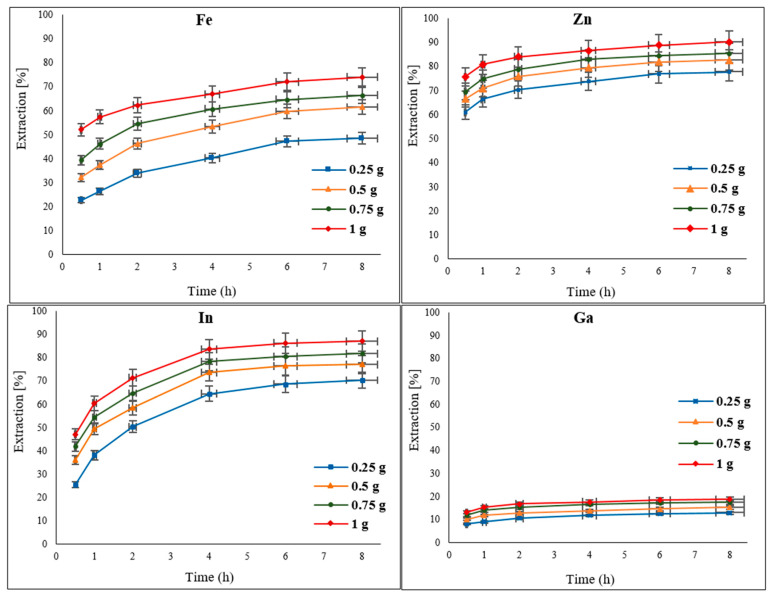
Effect of different concentrations of the oxidant Fe_2_(SO_4_)_3_ (0.25, 0.5, 0.75, and 1 g) on the extraction of Fe, Zn, In, and Ga from EAFD. Experimental conditions: 50 mL of 30% *v*/*v* [Bmim^+^HSO_4_^−^], S/L ratio of 1/20, stirring speed of 500 rpm, temperature of 65 °C, and total leaching time of 8 h.

**Figure 7 materials-15-08648-f007:**
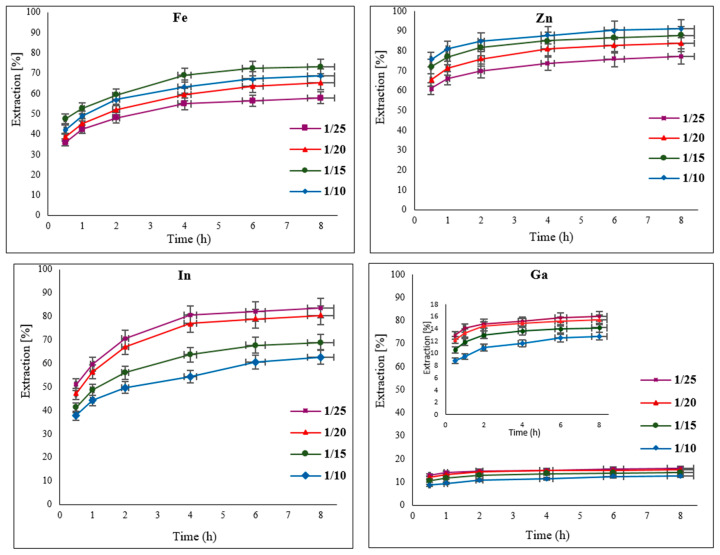
Effect of different solid-to-liquid (S/L) ratios (1/25, 1/20, 1/15, and 1/10) on the extraction of Fe, Zn, In, and Ga from EAFD. Experimental conditions: 50 mL of 30% *v*/*v* [Bmim^+^HSO_4_^−^], 1 g of oxidant Fe_2_(SO_4_)_3_, stirring speed of 500 rpm, temperature of 65 °C, total leaching time of 8 h.

**Figure 8 materials-15-08648-f008:**
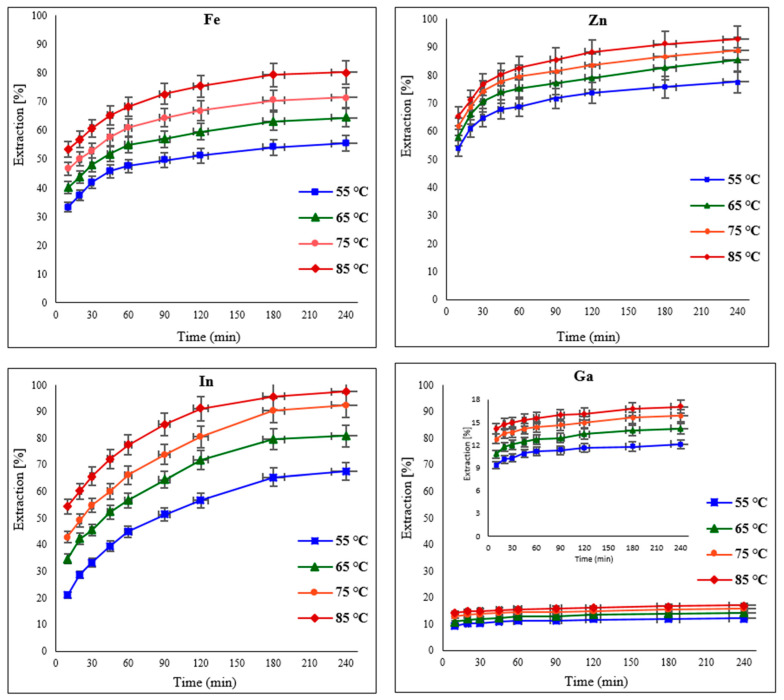
Effect of varying temperatures (55, 65, 75, and 85 °C) on the extraction of Fe, Zn, In, and Ga from EAFD. Experimental conditions: 100 mL of 30% *v*/*v* [Bmim^+^HSO_4_^−^], 1 g of oxidant Fe_2_(SO_4_)_3_, S/L ratio of 1/20, stirring speed of 500 rpm, and total leaching time of 4 h (240 min).

**Figure 9 materials-15-08648-f009:**
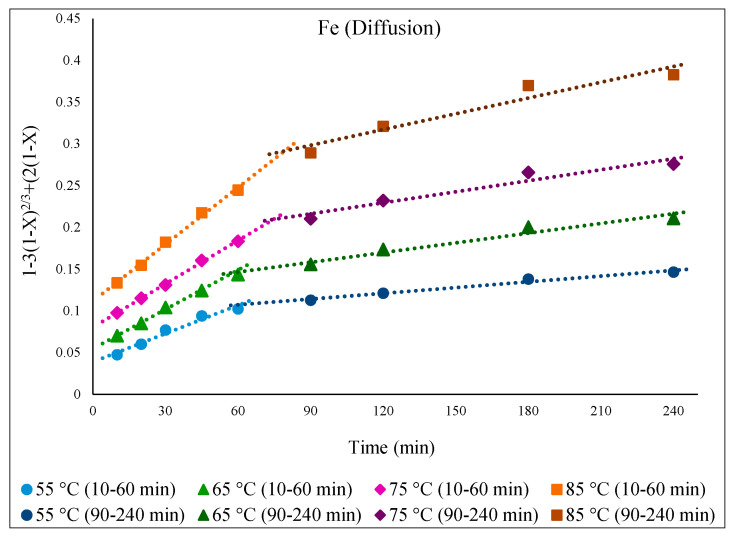
Graph of [1 − 3(1 − X)^2/3^ + 2(1 − X) = k.t] over time at different temperatures of 55–85 °C for the dissolution of Fe from EAFD in 30% *v*/*v* [Bmim^+^HSO_4_^−^] solution with 1 g of oxidant Fe_2_(SO_4_)_3_.

**Figure 10 materials-15-08648-f010:**
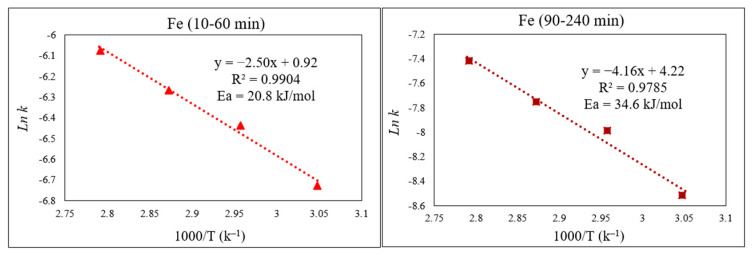
Arrhenius graph of *ln* k vs. 1000/T for the first part (10–60 min) and the second part (90–240 min) of Fe extraction.

**Figure 11 materials-15-08648-f011:**
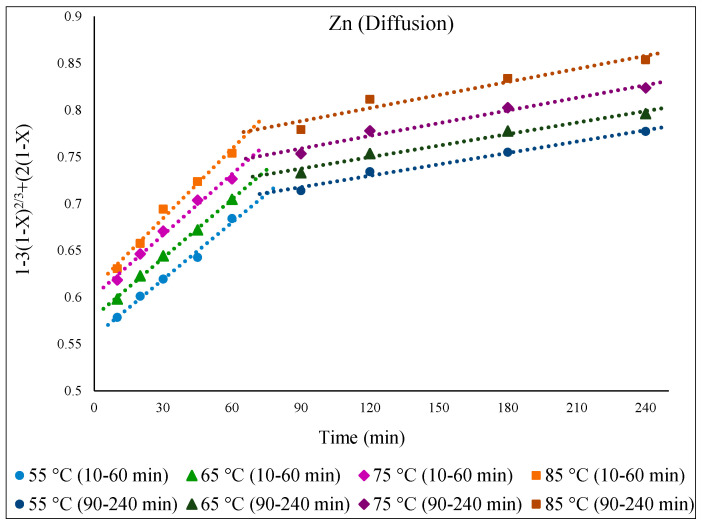
Plot of [1 − 3(1 − X)^2/3^ + 2(1 − X)= k.t] over time (min) showing progress at different temperatures of 55–85 °C for the dissolution of Zn from EAFD in 30% *v*/*v* [Bmim^+^HSO_4_^−^] solution with 1 g of oxidant Fe_2_(SO_4_)_3_.

**Figure 12 materials-15-08648-f012:**
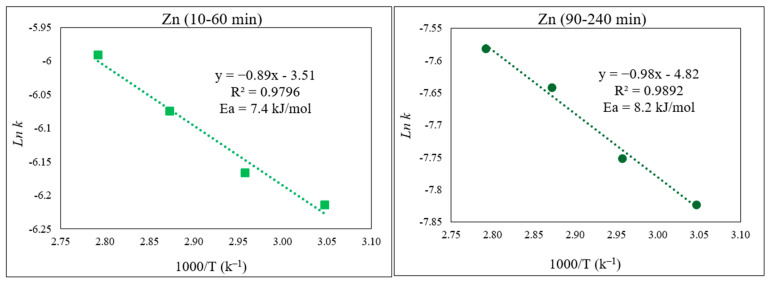
Arrhenius plot of *ln* k vs. 1000/T for Zn for the first part of the extraction (10–60 min) and the second part (90–240 min) of the extraction.

**Figure 13 materials-15-08648-f013:**
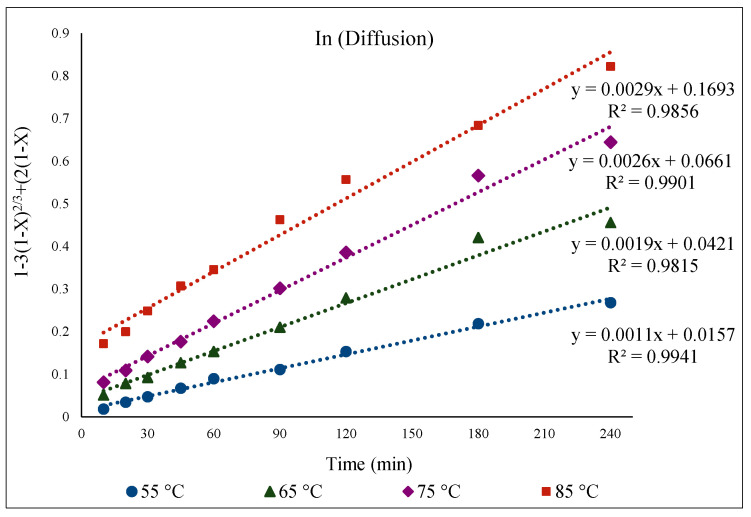
[1 − 3(1 − X)^2/3^ + 2(1 − X) = k.t] over time (min) for temperatures from 55 to 85 °C for In extraction in 30% *v*/*v* [Bmim^+^HSO_4_^−^] solution with 1 g of oxidant Fe_2_(SO_4_)_3_.

**Figure 14 materials-15-08648-f014:**
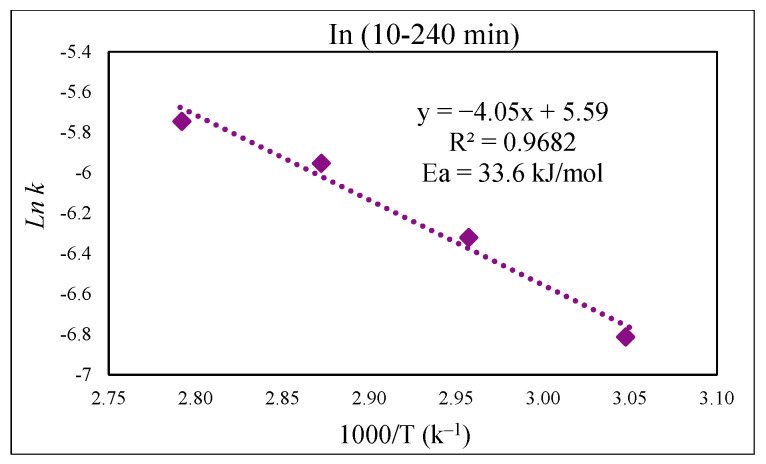
Arrhenius graph of *ln* k vs. 1000/T for In.

**Figure 15 materials-15-08648-f015:**
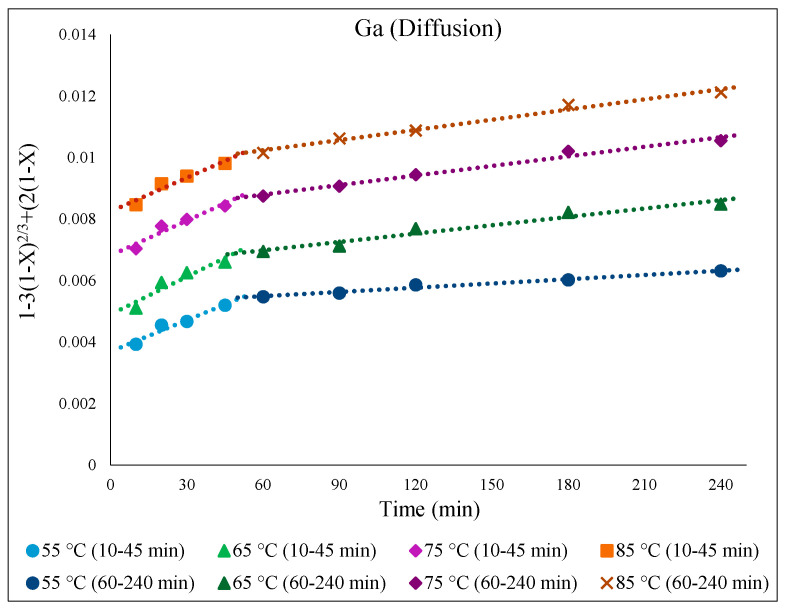
[1 − 3(1 − X)^2/3^ + 2(1 − X) = k.t] over time (min) for temperatures from 55 to 85 °C for Ga extraction in 30% *v*/*v* [Bmim^+^HSO_4_^−^] solution with 1 g of oxidant Fe_2_(SO_4_)_3_.

**Figure 16 materials-15-08648-f016:**
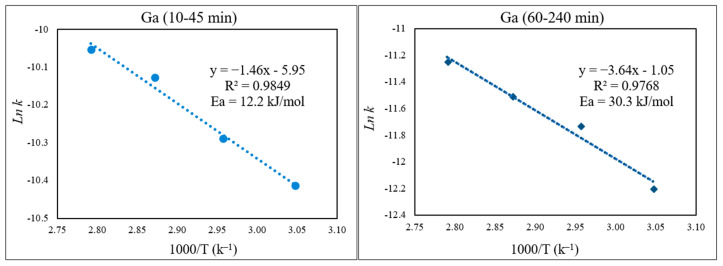
Arrhenius plot of *ln* k vs. 1000/T for Ga for the first part (10–45 min) and the second part (60–240 min) of the extraction.

**Figure 17 materials-15-08648-f017:**
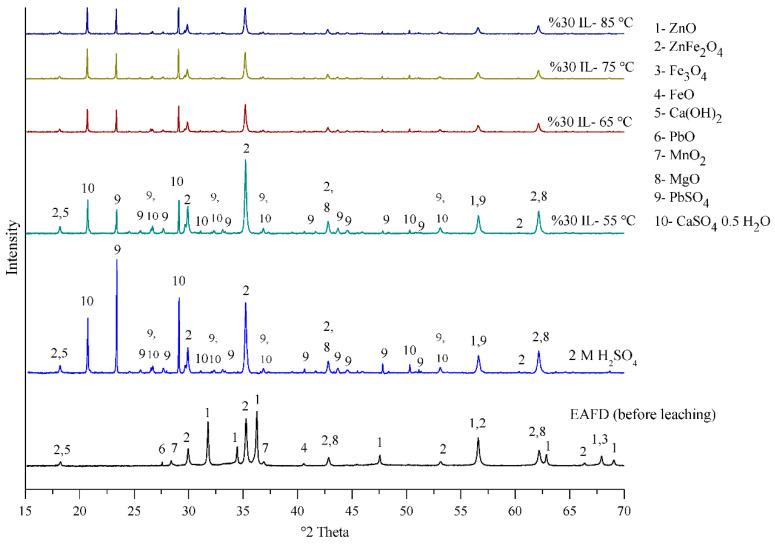
XRD pattern of EAFD before and after leaching in 30% *v*/*v* [Bmim^+^HSO_4_^−^] solution at different temperatures (55 to 85 °C) and in 2 M H_2_SO_4_ (85 °C) for comparison.

**Figure 18 materials-15-08648-f018:**
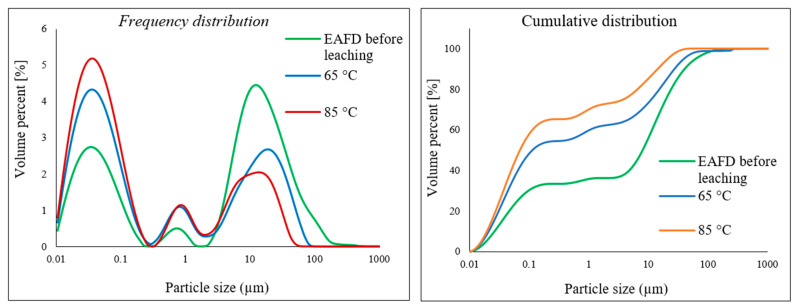
Comparing the PSD of the EAFD before leaching and the particles leached in 30% *v*/*v* [Bmim^+^HSO_4_^−^] solution at 65 and 85 °C.

**Figure 19 materials-15-08648-f019:**
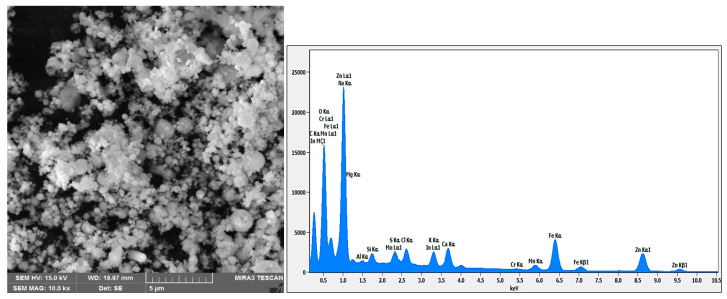
SEM-EDS image of EAFD before leaching, with the chemical composition and the coloured map of the elements in the examined area.

**Figure 20 materials-15-08648-f020:**
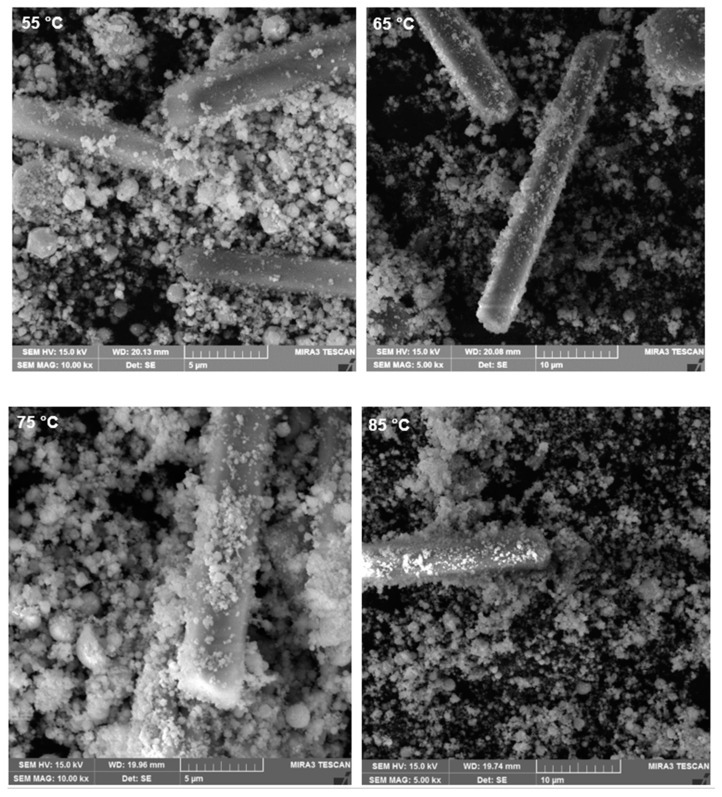
SEM images of the EAFD residue after leaching in 30% *v*/*v* [Bmim^+^HSO_4_^−^] solution at different temperatures (55 to 85 °C).

**Figure 21 materials-15-08648-f021:**
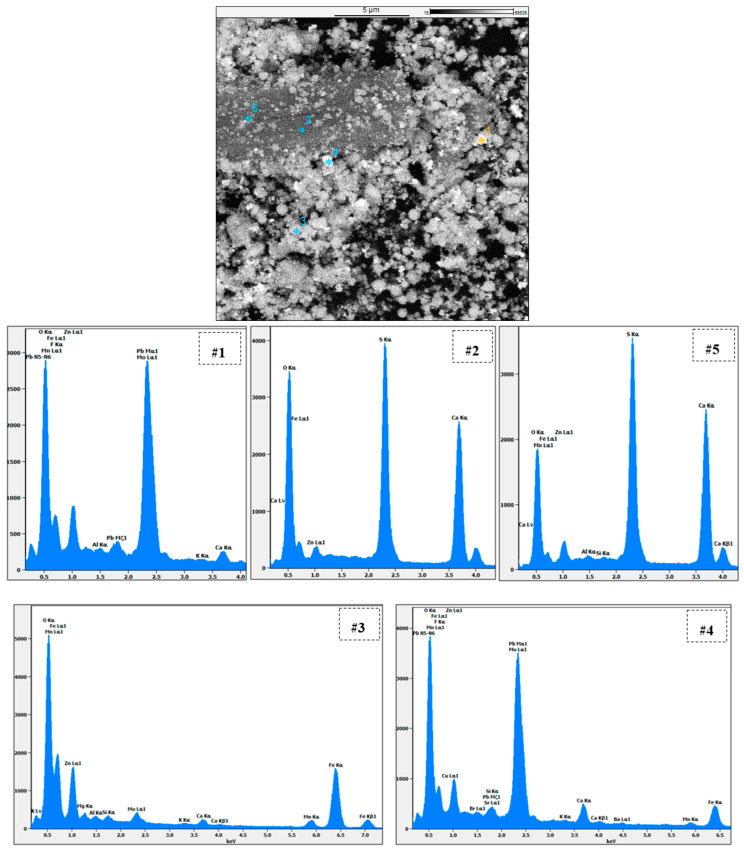
Point analysis with SEM-EDS of the EAFD leach residue in 30% *v*/*v* [Bmim^+^HSO_4_^−^] at 85 °C.

**Figure 22 materials-15-08648-f022:**
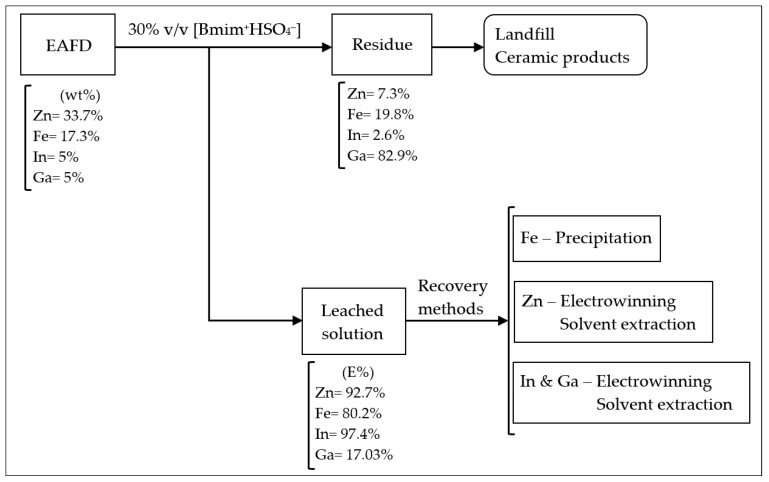
A scheme for the proposed treatment process(es) of EAFD leaching showing the residue and the leached solution.

**Table 1 materials-15-08648-t001:** Chemical analysis of the main elements in the EAFD sample (total acid digestion, measured by * AAS, ** ICP-OES, and *** S/C analyser) [14].

Element	Content [wt%]
Zn	33.2 *
Fe	17.9 *
Ca	3.6 **
Mn	2.5 *
K	2.4 **
Na	1.8 **
Pb	1.64 *
C	1.17 ***
S	1.05 **
Si	0.83 **
Mg	0.64 **
Al	0.36 **
Cr	0.23 **
Cu	0.20 **
Ni	0.03 **
V	0.02 **
P	0.02 **

**Table 2 materials-15-08648-t002:** The values for activation energy and extraction efficiency (%E) for Zn and Fe.

Reagent Used	E_a_ for Zn (kJ/mol)		%E Zn	E_a_ for Fe (kJ/mol)		%E Fe	Reference
	1st Part	2nd Part		1st Part	2nd Part		
1 M H_2_SO_4_	1.7	42.3	87%	35.6		79%	[6]
30% *v*/*v* [Bmim^+^HSO_4_^−^]	7.4	8.2	92.7%	20.8	34.6	80.2%	This research

**Table 3 materials-15-08648-t003:** The chemical composition of the elements in the examined area.

Element	Content [wt.%]
Zn	38.5
O	27.5
Fe	20.4
Ca	3.5
Mn	2.6
K	1.97
Mo	1.96
Cl	1.19
Si	1.14
S	0.41
Mg	0.39
Al	0.28
Cr	0.24
Total	100

**Table 4 materials-15-08648-t004:** Chemical composition of the selected points (1 to 5) on the EAFD residue leached in 30% *v*/*v* [Bmim^+^HSO_4_^−^] at 85 °C.

Area	Content [wt.%]													
	Zn	O	Fe	Pb	Mo	Mn	Mg	Ca	Cu	K	S	Si	F	Al
#1	5.6	31	15.7	27.7	11.0	1.8	---	1.5	---	0.22	3.2	---	1.9	0.33
#2	1.6	52.1	6.7	---	---	0.51	---	21.7	---	---	17.4	---	---	---
#3	11.2	36.4	42.9	---	2.8	3.2	1.01	0.1	---	0.13	---	0.7	---	0.5
#4	4.5	38.6	10.7	13.8	21.4	1.4	---	2.8	1.35	0.18	2.2	0.33	1.9	---
#5	5.1	40.3	8.5	---	---	1.1	---	25.1	---	---	19.7	0.11	---	0.19

## Data Availability

Not applicable.

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
