# Peer review of "A New Hydrometallurgical Process for Metal Extraction from Electric Arc Furnace Dust Using Ionic Liquids"

_materials, 2022, doi:10.3390/ma15238648_

Round 1
Reviewer 1 Report
1. Not clear if the EAFD has been analyzed after acid leaching or the previously results are given in Table 1.
2. Not clear if S and C are analyzed under the same procedure.
3. The obtained results are very important however I don't see refetrences on the research done in the same direction in China by Jiugang Hu et al in 2012-2013.
4. Comparison with traditional (Waelz process) technologies will also improve the article.
5. Nothing is mentioned about Zn and other metals extraction by leaching in the persence of strong magnetic field. See Russian Patent #261708C1.
Author Response
Dear Reviewer 1:
Thank you for your constructive comments. We endeavoured to addressed each of them thoroughly and revised the manuscript accordingly.
Please find the attached file, our response to your comments.
Thanks.
Samaneh Teimouri

Reviewer 2 Report
I congratulate the authors for their work and excellent presentation.
I attach some suggestions and doubts about the work:
1. Why was the ionic liquid [Bmim+] selected, what advantages does it have over others?
2. What are the advantages and disadvantages of the proposed method with respect to the thermal method or the use of conventional acids or alkalis?
3. Complete the extraction and dissolution reaction, including the presence of the oxidizing agent.
4. Why was Fe(SO4)3 selected instead of H2O2 since the results are very similar using both oxidants? the reason was cost, handling, safety? what are the advantages of using Fe(SO4)3 over H2O2?
5. How many repetitions did you make of your experiments?
6. Page 10, lines 3-6. The explanation given in relation to obtaining a lower extraction with a higher mass of EAFD, may be due rather to the fact that the ionic liquid is the limiting reagent in this reaction?
7. Why was the shrinking core model chosen to explain the extraction kinetics of these metals by [Bmim+HSO4-] from EAFDs?
8. In Figure 15, how the two stages of the extraction process are explained, which mineral phases would then be involved for Ga?
9. Is the intensity scale in Figure 17 quantitative? Were the areas under the curve of each diffractogram determined or was any adjustment made to determine the concentrations of the phases?
10. In Figure 17, "before" is missing an "e".
11. The low extraction efficiencies (17%) obtained for Ga, allow us to assert that this method is not exactly the ideal one to extract Ga. This should be clarified in the paper. It is a method more recommended for In or Zn, but I do not consider it to be for Ga.
Author Response
Dear Reviewer 2:
We appreciate your encouraging and kind words on this research. Thanks for your constructive comments that helped to improve this work. We went through all your comments and addressed each of them thoroughly and revised the manuscript to improve it.
Please find the attached file, our response to your comments.
Thanks.
Best Regards,
Samaneh Teimouri

Reviewer 3 Report
Manuscript ID: materials-2048416
Title: A new hydrometallurgical process for metal extraction from electric arc furnace dust using ionic liquids
Authors: Samaneh Teimouri et al.
The authors have done an impressive job. This is a very well-written article that contains detailed information. The introduction well describes all the latest methods of using ionic liquids for the extraction of zinc, iron, and rare-earth metals. The method has been wonderful. There are no questions about how to repeat these experiments. Results look convincing, a huge number of experiments have been carried out. I really think this is an excellent article. I could accept right away, but I ask the authors to discuss minor changes to this article. Article might get even better. Here are my wishes and questions:
Figure 4-8. Authors should add the error bars for experimental points in figures.
· Authors can add information about yield (%) of leaching residue after leaching tests. I cannot find this information in the article.
Figure 17. I was found the Chinese characters on XRD patterns names in figures. It must be removed. Improve the “CaSO4.0.5H2O” to “CaSO4·0.5·H2O”.
Section 3.6. Change The functions from this model considered the surface chemical reaction [(1 – (1 – X)1/3 = k.t], diffusion through the product [1 – 3(1–X)2/3 + 2(1–X) = k.t], to The functions from this model considered the surface chemical reaction [(1 – (1 – X)1/3 = kt], diffusion through the product [1 – 3(1–X)2/3 + 2(1–X) = kt]
· The article shows that gallium is very poorly extraction, what is the reason for this? What mineral contains gallium? How can authors increase Ga extraction? The authors can use TEM with mapping to search for gallium in dust minerals.
· There is also a question about the calculation of kinetics. Why the authors do not use point “0”. Without using this point, the calculation is not entirely correct. Authors can read more information here: Levenspiel O. Chemical Reaction Engineering. https://the-seventh-dimension.com/images/textlev/LEVENSPIEL%20Chemical%20reaction%20engineering-ch1-ch2.pdf
I think the authors can add this point for a more accurate calculation of the activation energy in the first 60 min of leaching process.
Authors should add the chemical compositions of residue (wt.%) and liquor (g/L) after leaching.
· Authors can add discussion of the future methods to separate the Zn and Fe from In and Ga from the liquor, and How is it planned to use the solid residue after leaching?
· I think if the authors added at the end of the article a technological scheme with optimum technological parameters, the contents of elements (Zn, Fe, In, Ga) in raw dust, residue and solution, and further operations, this would improve the article.
Author Response
Dear Reviewer:
Thank you for your kind and encouraging words, and the constructive comments that helped to improve this work. We addressed each of your comments and revised the manuscript based on them to improve it.
Please find the attached file, our response to your comments.
Thanks.
Best regards,
Samaneh Teimouri

Round 2
Reviewer 3 Report
Article can be accepted in present form.